# RECLIP: Resource-efficient CLIP by Training with Small Images

Runze Li*   Dahun Kim †   Bir Bhanu*   Weicheng Kuo †

UC Riverside*   Google Deepmind†

Reviewed on OpenReview: `https://openreview.net/forum?id=Ufc5cWhHko`

## Abstract

We present RECLIP (Resource-efficient CLIP), a simple method that minimizes computational resource footprint for CLIP (Contrastive Language Image Pretraining). Inspired by the notion of coarse-to-fine in computer vision, we leverage small images to learn from large-scale language supervision efficiently, and finetune the model with high-resolution data in the end. Since the complexity of the vision transformer heavily depends on input image size, our approach significantly reduces the training resource requirements both in theory and in practice. Using the same batch size and training epoch, RECLIP achieves highly competitive zero-shot classification and image-text retrieval accuracy with 6 to $8\times$ less computational resources and 7 to $9\times$ fewer FLOPs than the baseline. Compared to the state-of-the-art contrastive learning methods, RECLIP demonstrates 5 to $59\times$ training resource savings while maintaining highly competitive zero-shot classification and retrieval performance. Finally, RECLIP matches the state of the art in transfer learning to open-vocabulary detection tasks, achieving 32 AP$r$ on LVIS. We hope this work will pave the path for the broader research community to explore language supervised pretraining in resource-friendly settings.

## 1 Introduction

Representation learning is a foundational problem in computer vision and machine intelligence. Effective image representation can benefit a myriad of downstream tasks, including but not limited to image classification, object detection, semantic segmentation, and 3D scene understanding. In the past decade, the community has witnessed the rise of supervised learning (Deng et al., 2009; Sun et al., 2017), then self-supervised learning (Chen et al., 2020; He et al., 2020; Bao et al., 2022), and most recently language-supervised learning (Radford et al., 2021; Jia et al., 2021; Yu et al., 2022). Language-supervised representation gains much traction for its exceptional versatility. It exhibits outstanding performance in zero-shot classification (Radford et al., 2021), linear probing (Radford et al., 2021; Yu et al., 2022), few-shot learning (Zhou et al., 2022), full finetuning (Dong et al., 2022a), and finds great applications in text-guided image generation (Ramesh et al., 2021). Much like the role of supervised pretraining (Deng et al., 2009) before, language-supervised pretraining has emerged as a simple yet powerful methodology for representation learning today.

Traditional supervised learning uses a predetermined set of labels, and is effective across a wide range of data and computational resources. In contrast, natural language offers richer learning signals such as object categories or instances, named-entities, descriptions, actions, and their relations at multiple levels of granularity. Unfortunately, this rich supervision also leads to a higher level of noise in the data, where many image-text pairs have only loose connections. To address this noise, data and computational scaling have proven to be highly effective and necessary. For example, training CLIP models require $\sim$3k V100-GPU-days, and likewise CoCa requires $\sim$23k TPU-v4-core-days. Apart from the lengthy training time, the large batch requirement of contrastive learning recipes also demand substantial amount of device memory at all times. These factors limit the research of language supervised learning to institutions with high-end infrastructure, and hinder the exploration by the broader community.

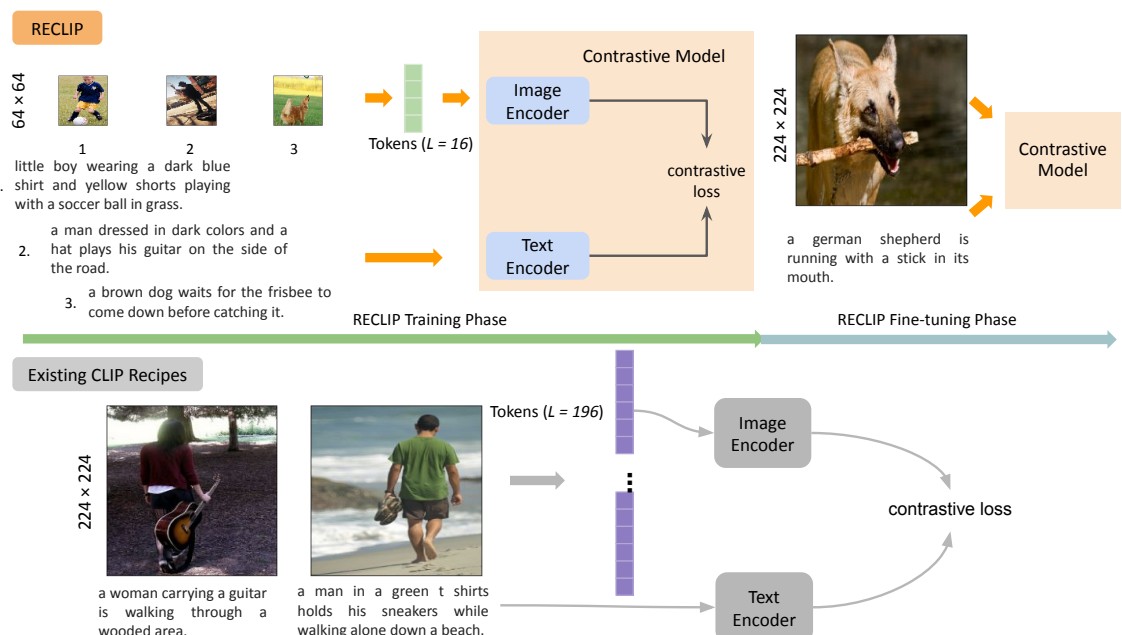

Figure 1: Top: Resource-efficient CLIP (RECLIP) training pipeline. Bottom: existing CLIP training methods. RE-CLIP leverages small images for the main training phase which significantly reduces computational resource requirements through much shorter image sequence length.

Thus, improving efficiency of contrastive training has drawn substantial research interest. For example, Zhai et al. (2022) precomputes the image features by a pretrained classification model to reduce the training cost. Zhai et al. (2023) utilizes sigmoid loss to avoid the use of all-gather operation and improves learning with a smaller batch size. Moreover, Yao et al. (2021) leverages masked images to speed up contrastive learning. The community have also explored smaller batch sizes (Dong et al., 2022b) or curated academic datasets (Li et al., 2022a; Lei et al., 2022) for contrastive learning. However, it is not clear how well the findings in smaller batch and data size settings generalize to larger batch and data size.

We present RECLIP (Resource-efficient CLIP), a simple method designed to make CLIP more affordable and reproducible for the community (see Fig. 1). Consider images 1-3 in the top left of Fig. 1. Humans can effortlessly match the images with the corresponding texts below them, e.g. "a boy is playing a soccer ball in grass" matching image 1. Although the images are only of size $64 \times 64$, they contain adequate amount of visual information for pairing with texts. Our main insight is to train on small images during the main training phase, and finetune the model with high-resolution images for a short schedule in the end. Intuitively speaking, our approach re-introduces the idea of "coarse-to-fine" from classical computer vision to contrastive learning, whereby pretraining incorporates high-level information from small images and finetuning enables the model to refocus its attention on the important details. There is no need for multi-view supervisions (Li et al., 2022a; Yao et al., 2021), feature distillation (Lei et al., 2022), other contrastive losses (Zhai et al., 2023), pretrained classifiers (Zhai et al., 2022), or image masking (Li et al., 2022b). Surprisingly, RECLIP achieves highly competitive zero-shot classification and retrieval performance using $64 \times 64$ images, which significantly reduces computational resource usage. We attribute this to the complexity of image tower being quartic with respect to the image size (see Eqn. 4).

In addition, RECLIP demonstrates the efficiency and effectiveness of using short sequence length for image language representation learning. Existing image-text pretraining methods typically use long sequence lengths, e.g. 441 (Radford et al., 2021) or 784 (Yu et al., 2022) to achieve strong downstream zero-shot transfers. Long sequence image encoding has been validated to benefit image classification (Beyer et al., 2022) and object detection (Chen et al., 2022a) with vision transformers. Hu et al. (2022) find the sequence length is a key factor for masked image representation learning. Different from these methods that advocate for long sequence length, RECLIP demonstrates that using only **16** tokens for the image encoding is sufficient for the main training phase, and can achieve highly competitive zero-shot transfer capabilities via a short high-resolution finetuning schedule. Interestingly, our image sequence length is 4 to 5× shorter than the *text* sequence lengths of popular recipes e.g. 76 (Radford et al., 2021) or 64 (Yu et al., 2022).

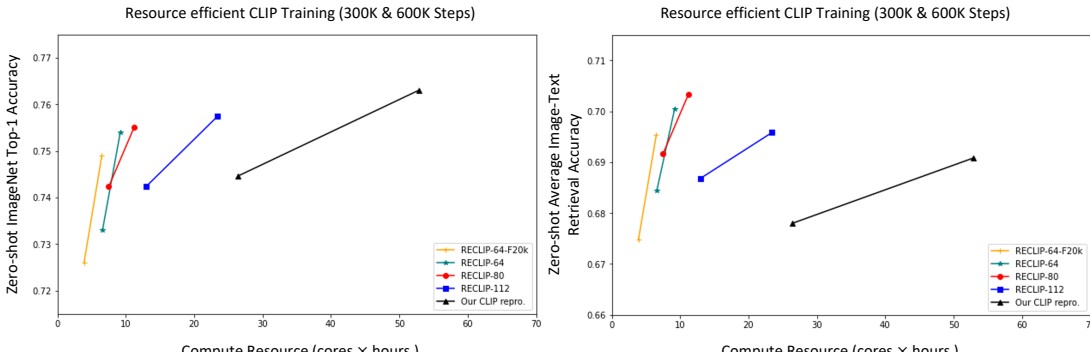

Figure 2: Zero-shot accuracy *vs.* compute resource in cores×hours trade-off. RECLIP-X: RECLIP training for 300k and 600k steps with image size $X$ where $X = 64, 80, 112$. RECLIP-64-F20k: RECLIP-64 finetuned for 20k steps. Our CLIP repro.: our reproduction of CLIP (Radford et al., 2021). Zero-shot image-text retrieval results are averaged from image-to-text and text-to-image Recall@1 on two benchmark datasets, Flickr30K (Plummer et al., 2015) and MSCOCO (Chen et al., 2015). RECLIP consumes significantly less compute resource and is more accurate on zero-shot image-text retrieval and highly competitive classification results on ImageNet-1K validation set.

In Fig. 2, we present zero-shot classification and retrieval performance, and resource costs in cores×hours of training RECLIP models and the baseline model for short and long schedules. Experiments show that, using the same batch size and training steps, RECLIP reduces the computation resources by 6 to 8× and largely preserves the classification and retrieval accuracy. When comparing to state-of-the-art (SOTA) methods, RECLIP significantly saves resource usage by 5 to 59× and shows highly competitive zero-shot classification and retrieval accuracy. Apart from image-level tasks, we explore transfer learning of RECLIP to open-vocabulary detection tasks (Gu et al., 2022), which typically requires high-resolution images for small object recognition. Surprisingly, RECLIP achieves 32 $AP_r$, matching the state of the art performance of RO-ViT (Kim et al., 2023) on LVIS benchmark. This demonstrates the potential of RECLIP for region and pixel-level tasks beyond image-level understanding. In summary, our contributions are:

- We present a new language image pretraining methodology, Resource-efficient CLIP (RECLIP) to minimize computational resource requirements.

- We leverage small images for the main contrastive learning phase to enable the model to be trained with language supervisions fast and then finetune the model on high-resolution data with a short schedule in the end.

- RECLIP significantly saves compute resource, reduces FLOPs and achieves highly competitive performance on both zero-shot classification and image-text retrieval benchmarks.

- RECLIP matches the state of the art in open-vocabulary detection with much less training resources.

We believe RECLIP could enable the broader research community to explore and understand language supervised pretraining in a more resource friendly setting.

## 2 Related Work

### 2.1 Learning with Low-Resolution Images

Deep learning techniques have been utilized on a wide-variety of computer vision tasks, e.g. visual recognition (He et al., 2016; Dosovitskiy et al., 2021), video analysis (Tran et al., 2018), images generations (Ramesh et al., 2021), etc. Most of existing work follow the standard training and testing paradigms to exploit very deep models by using images with the fixed resolution, e.g. $224 \times 224$. This setting has been one of fundamental standards for various computer vision tasks. However, an increasing number of studies have been conducted to investigate to train deep learning models with low-resolution data. Touvron et al. (2019) have observed significant discrepancy on image sizes

caused by augmentation methods during the train and test period, and further validated the effectiveness of using lower resolution images for training than testing. Driven by the needs for specific tasks, e.g. face recognition, surveillance images analysis, etc., Singh et al. (2019; 2022) and Huang et al. (2022) study learning with low resolution images and generally focus on using high resolution images as auxiliary data to help to train models with low resolution data, which causes difficulties to generalize on broader visual recognition tasks. For video understanding, Wu et al. (2020) propose to use variable mini-batch shapes with different spatial-temporal resolutions for training deep video models and obtain optimal performance and time trade-offs. With recent advances of vision transformers (Dosovitskiy et al., 2021; He et al., 2022), Guo et al. (2022) speedup image pretraining by using masked image modelling with low resolution data. Liu et al. (2022) introduce a a log-spaced continuous position bias for pretraining vision models by using smaller images and transfer to high-resolution localization tasks.

## 2.2 Language-supervised Learning

Due to the natural co-occurrence of image and language data on the web, language-supervised learning has become a highly effective and scalable representation learning methodology. Researchers have explored a variety of paired image-text data such as image tags (Chen & Gupta, 2015; Divvala et al., 2014; Joulin et al., 2016), captions (Desai & Johnson, 2021; Sariyildiz et al., 2020; Wang et al., 2009; Sharma et al., 2018), alt-texts (Jia et al., 2021; Schuhmann et al., 2021), image search queries (Radford et al., 2021), page title (Chen et al., 2022b), or a combination of these sources (Chen et al., 2022b). From a modeling perspective, contrastive learning is particularly suitable for recognition and retrieval tasks, because of its simplicity and versatility. However, the high requirements of computational resources have limited the research from the broader community.

To fully leverage capabilities of vision and language pretraining, large batch size (e.g. 16k (Jia et al., 2021), 32k (Radford et al., 2021; Yao et al., 2021), or 64k (Yu et al., 2022)) and web image text data have been adopted widely. This requires a large amount of computational resources which many academic institutions and industry labs cannot afford. To address such limitation, Zhai et al. (2022) proposes to precompute the image features with frozen classifier backbone, while Zhai et al. (2023) proposes sigmoid loss which better supports small batch training. In addition, masked image learning (Yao et al., 2021), multi-views data augmentations (Li et al., 2022a; Yao et al., 2021), knowledge distillations (Lei et al., 2022) and masked self-distillation Dong et al. (2022b) have been proposed. Since many of these methods are trained and evaluated on smaller scale/data, it is unclear how well they may scale up to larger batch and data. For example, Weers et al. (2023) shows that the advantage of contrastive learning approaches on smaller scales may not always hold at larger scales. In contrast, we propose a simple and novel recipe for language image pretraining, which (1) significantly reduces computation resource requirements and (2) works well on a large-scale web dataset Chen et al. (2022b) with minimal change to the established CLIP recipe (Radford et al., 2021).

# 3 Method

## 3.1 Preliminaries

**Contrastive Language Image Pretraining.** Following existing works (Radford et al., 2021; Yu et al., 2022), we utilize a transformer-based contrastive model which consists of an image encoder and a text encoder. The image and text encoders are trained to output image-level representation and sentence-level representations respectively. The image embeddings $\{p\}$ and text embeddings $\{q\}$ are obtained by global average pooling at the last layers of image and text encoders. The cosine similarity of the embeddings in batch $B$, scaled by a learnable temperature $\tau$ are the input to the InfoNCE loss (Oord et al., 2018; Radford et al., 2021). The image and text contrastive loss is obtained by $L_{con} = (L_{\text{I2T}} + L_{\text{T2I}})/2$, with:

$$L_{\text{I2T}} = -\frac{1}{B}\sum_{i=1}^{B}\log\left(\frac{\exp(p_i q_i/\tau)}{\sum_{j=1}^{B}\exp(p_i q_j/\tau)}\right). \tag{1}$$

$$L_{\text{T2I}} = -\frac{1}{B}\sum_{i=1}^{B}\log\left(\frac{\exp(q_i p_i/\tau)}{\sum_{j=1}^{B}\exp(q_i p_j/\tau)}\right). \tag{2}$$

where $i, j$ are indexes within the batch. This loss is optimized to learn both the image and language representation in the dual-encoder model.

## 3.2 Resource-efficient CLIP

At a high-level, our method utilizes small images to reduce computation and leverage a brief finetuning stage at the end of training to adapt for high-resolution inference. Intuitively, the use of smaller images presents a trade-off between how much detail we encode per example and how many samples we process per unit of computation resource. Fig. 1 shows the RECLIP training pipeline on the top. There are two phases: low-resolution main training, and high-resolution finetuning. In the first phase, we leverage small images which contain sufficient visual concepts with paired texts as the input to the image and text encoders. By using an image size of $64$ and a text length of $16$, RECLIP processes the training data significantly faster than existing methods. In the second phase, we finetune the model for a short cycle on high-resolution data to provide valuable image details, which largely enhances the representation quality of the model. Below we delve deeper into specific aspects of RECLIP design.

**Structure preservation by learning from small images.** In Fig. 1, we observe that small images can preserve visual structure and contain sufficient concepts well. For instance, human can easily tell the object, "a dog", in the third image and associate the image with the text of "a brown dog waits for ...", and this is a fundamental principle for our RECLIP to leverage small images for the main language-supervised pretraining. Because down-sampling is a structure-preserving operation i.e. global appearance remains similar, we are able to reduce the token length aggressively without compromising the performance of the model. This is different from other techniques to reduce the sequence length (e.g. random masking) where the global appearance may change significantly with reduced sequence lengths. Additional visualization presents a comparison between various image resolutions and sheds light on how small images effectively preserve visual appearance (see Fig. 3).

**Training complexity with small images.** The computation cost of contrastive learning mostly depends on the cost of processing images (Radford et al., 2021; Li et al., 2022b; Yu et al., 2022), partly because the image encoder is typically heavier than the text encoder, partly because the image token length tends to be greater than that of text tokens. Below we provide theoretical analysis to understand the efficiency of using small images.

Let the number of tokens from the image encoder be:

$$N = hw/p^2 \tag{3}$$

, where $h/w$ are height/widths of the image, and $p$ is the patch size. If we replace $h$ by $H/r$ and $w$ by $W/r$, where $H/W$ are the original image height and widths, and $r$ is the down-sampling factor. The computation complexity $C$ of the image encoder of a batch is given by :

$$C = O(BN^2) = O(\frac{BH^2W^2}{p^2r^4}) \tag{4}$$

, where $B$ is the batch size. When $B, H, W, p$ are held constant, we have:

$$C = O(\frac{1}{r^4}) \tag{5}$$

This shows that reducing the image size is very effective in reducing computation complexity to the inverse power of up to 4. Since image encoder is the computation bottleneck in existing CLIP recipes (Radford et al., 2021; Li et al., 2022b; Zhai et al., 2022; Yu et al., 2022), RECLIP reduces the image sequence length to **16** by using an image size of 64, which makes our image token length the same as our own text token length, and much shorter than those of aforementioned methods.

The above complexity analysis C is calculated based on the core operations self-attention layers in transformers. However, empirically the complexity of a transformer may not be dominated by the self-attention layers, the fully connected layers also play an important role. GPT-3 (Brown et al., 2020) paper have provided computation analysis of their language models, where the computation cost is estimated as $O(N)$, linear with the sequence length. Thus, we discussed the lower-bound of the complexity $C_{lb}$ of RECLIP in equation 6. Using the notation of equation 4 and equation 5, we have

$$C_{lb} = O(BN) = O(\frac{BHW}{pr^2}), \tag{6}$$

During the training, the B, H, W, p are normally constant, so equation 6 can be simplified as:

$$C = O(\frac{1}{r^2}). \qquad (7)$$

Compared to equation 5, we observe that the computation savings in practice may be somewhere between $O(\frac{1}{r^2})$ and $O(\frac{1}{r^4})$. This analysis shows that changing $r$ is very effective regardless of the compute estimation techniques.

**Constant batch size.**    Batch size is a critical factor in contrastive learning (Radford et al., 2021; Pham et al., 2021; Li et al., 2022b; Chen et al., 2022a) and larger batch has consistently yielded improvement. In Equation 4, the complexity changes linearly with batch size $B$. Observing that reduced batch size tends to hurt representation quality, we keep the batch size constant to save both computation and memory use by reducing image size only.

**High-resolution finetuning.**    We perform high resolution finetuning after the main low-resolution training. Intuitively speaking, the model has acquired a high-level understanding of the images and texts through the main training phase. We improve its representation further by providing more detailed visual information through a short high-resolution finetuning process. The images used for high-resolution training are the same as those for the low-res training, except that we remove the downsampling to preserve the rich visual details.

Care is taken to initialize the positional embeddings from low-res pretraining to high-res finetuning. We up-sample the positional embedding weights from the low dimension (e.g. 4x4 for $h = w = 64$) to the dimension of high-resolution positional embeddings (e.g. 14x14 for $h = w = 224$) for a given patch size $p = 16$. Compared to up-sampling the low-res positional embeddings without increasing the amount of weights, we found this weight up-sampling beneficial because the positional embeddings have higher capacity to adapt with more detailed spatial representation. We use trainable positional embedding throughout the paper following existing works (Radford et al., 2021; Dosovitskiy et al., 2021; Yu et al., 2022).

**Network architecture.**    We use the ViT-Large backbone as image encoder by default unless noted otherwise. The ViT-Large is a vision transformer which consists of 24 multi-head self-attention layers with 16 heads and the width dimension of 1024. The patch size is fixed at 16 following common practice. Although we focus on ViT architecture in this study, RECLIP involves only changing the input size, and can potentially support other network architectures as well (Vaswani et al., 2017; Dosovitskiy et al., 2021; He et al., 2016; Liu et al., 2021; Tolstikhin et al., 2021). Our text encoder follows the same transformer design as previous works (Radford et al., 2021; Yu et al., 2022). The text encoder consists of 12 multi-head self-attention layers with 12 heads and the width dimension of 1024.

**Implementation details.**    We use a starting learning rate of 0.001, and train for 250k and 550k steps with linear LR decay using an Adafactor optimizer. We set weight decay to 0.01 and batch size to 16384. The batch size is chosen to be a multiple of 1024 and the model feature dimension (e.g. 4096) a multiple of 128, so that TPU padding would not occur on the sequence dimension. A short LR warmup of 2500 steps is used. Our high-resolution finetuning schedule starts with a learning rate of $10^{-4}$ with 5000 steps LR warmup, and decays linearly over a total schedule of 20k or 50k iterations. We use an image size of 224 or 448 for finetuning. We use the English subset of the WebLI dataset (Chen et al., 2022b) for training. Our training is run on TPU-v3 infrastructure. Compared to general-purpose GPU devices, TPUs are specifically designed for large matrix operations commonly used in neural networks. Each TPU v3 device has 16GB high-bandwidth memory per core, which is comparable to that of a V100 and suitable for synchronous large-scale training. For zero-shot image classification, we use the same text prompts as Radford et al. (2021).

## 4  Experiments

### 4.1  Main Results

**Zero-shot image-text retrieval and image classification.**    Following existing works (Radford et al., 2021; Li et al., 2022b; Yu et al., 2022), we evaluate RECLIP on zero-shot image and text retrieval on Flickr30K (Plummer et al., 2015) and MSCOCO (Chen et al., 2015) test sets, and zero-shot image classification on ImageNet (Deng et al., 2009), ImageNet-A (Hendrycks et al., 2021b), ImageNet-R (Hendrycks et al., 2021a), ImageNet-V2 (Recht et al., 2019) and ImageNet-Sketch (Wang et al., 2019) datasets. we take each image and text to the corresponding encoder to obtain embeddings for all image and text pairs. Then we calculate the the cosine similarity scores for the retrieval, and use the

Table 1: Zero-shot image-text retrieval, image classification results. CLIP*: The original CLIP model (Radford et al., 2021) is marked in gray. The resource use is converted to TPU-v3 core-hours per Li et al. (2022b). CLIP, our repro.: our reproduced CLIP. RECLIP-$X$: RECLIP trained with image size $X$ where $X = 64, 80, 112$. RECLIP-64-F20K: RECLIP-64 finetuned for a shorter schedule of 20k steps. Best results are **bolded**.

| Method | Training steps | Cores ×hours | Zero-shot Retrieval | | | | Zero-shot INet Classification | | | | |
| | | | Flickr30 (1K test set) | | MSCOCO (5K test set) | | INet | INet-A | INet-R | INet-V2 | INet-Sketch |
| | | | I2T R@1 | T2I R@1 | I2T R@1 | T2I R@1 | | | | | |
| CLIP* (Radford et al., 2021) | - | 120.0K | 88.0 | 68.7 | 58.4 | 37.8 | 76.2 | 77.2 | 88.9 | 70.1 | 60.2 |
| CLIP, our repro. | 300k | 26.4K | 89.3 | 75.4 | 61.3 | 45.1 | **74.5** | 54.4 | **88.9** | **67.7** | **64.5** |
| **RECLIP-112** | 300k | 13.1K | 90.0 | 76.6 | **63.1** | 45.0 | 74.2 | 55.4 | 87.8 | 67.2 | 63.2 |
| **RECLIP-80** | 300k | 7.5K | **91.0** | **77.1** | 62.8 | **45.7** | 74.3 | **56.7** | 87.8 | 67.2 | 62.9 |
| **RECLIP-64** | 300k | 6.6K | 89.4 | 77.0 | 62.2 | 45.2 | 73.3 | 53.7 | 86.3 | 66.2 | 61.6 |
| **RECLIP-64-F20K** | 270k | **3.9K** | 88.5 | 76.1 | 60.8 | 44.5 | 72.6 | 51.7 | 85.3 | 65.3 | 60.6 |
| CLIP, our repro. | 600k | 52.8K | 89.3 | 76.9 | 63.3 | 46.8 | **76.4** | 60.2 | **90.9** | **70.1** | **66.4** |
| **RECLIP-112** | 600k | 23.4K | 90.6 | 77.6 | 63.6 | 46.5 | 75.8 | 58.8 | 89.3 | 69.1 | 65.2 |
| **RECLIP-80** | 600k | 11.2K | **91.3** | **78.2** | **64.6** | **47.2** | 75.8 | 60.3 | 89.0 | 69.2 | 64.6 |
| **RECLIP-64** | 600k | 9.2K | 91.0 | 78.1 | 64.2 | 46.9 | 75.4 | **60.9** | 88.8 | 68.9 | 64.5 |
| **RECLIP-64-F20K** | 570k | **6.5K** | 91.0 | 77.1 | 63.6 | 46.2 | 74.9 | 58.6 | 88.2 | 68.4 | 63.5 |

aligned image and text embeddings to perform zero-shot image classification by matching images with label names without fine-tuning.

Table 1 presents the results of RECLIP on this benchmark, where the baseline is our own reproduced version of CLIP. Our baseline model trains on the WebLI dataset with the images of $224 \times 224$ for 300k and 600k steps. The original CLIP (Radford et al., 2021) model and trains on their own dataset with the image size of $336 \times 336$, which is marked in gray. RECLIP uses small images for the main training phase and finetune the model with the images of $224 \times 224$ for 20k or 50k steps.

For long-schedule training of 600k steps, RECLIP-64 significantly reduces compute use by $\sim$ **6** times from 52.8K to **9.2K** in cores× hours, which saves $\sim$ **80%** compute resource, and it outperforms the baseline model by +3.9 on Flickr and MSCOCO retrieval. RECLIP-64-F20K, which finetunes the model for only 20k steps with high-resolution images, further reduces the computation use by $\sim$ **8×** to **6.5K** and improves retrieval performance by +1.8. On zero-shot image classification, RECLIP-64 achieves 75.4 and RECLIP-64-F20K achieves 74.9 of the top-1 accuracy, which is very competitive with the baseline method. RECLIP-64 reduces the token length for the image encoding from 196 to **16** during the main training phase, which is a key factor for resource savings. Overall, RECLIP-64 shows attractive trade-offs between the resource use and zero-shot retrieval and image classification performance.

We also train RECLIP with the image size of $80 \times 80$. Comparing to the baseline method which consumes 52.8K in cores×hours, our RECLIP-80 remarkably reduces resource usage by $\sim$ **5** times to **11.2K**. RECLIP-80 improves retrieval results by +5.0 on Flickr30K and MSCOCO test sets, and achieves highly competitive zero-shot image classification performance of 75.8. Specifically, taking INet-A as an example, RECLIP-80 outperforms the baseline method for both 300k and 600k training steps. For short training schedule with 300 steps, RECLIP-80 requires only **7.5K** in cores×hours which is $\sim$ **4×** less than the baseline model.

Table 2: Comparisons of GFLOPs between RECLIP and the baseline model during the RECLIP training.

| Models | GFLOPs |
| --- | --- |
| CLIP, our repro. | 71.4 |
| **RECLIP-112** | 24.8 |
| **RECLIP-80** | 10.1 |
| **RECLIP-64** | 7.3 |

**GFLOPS.** We compare GFLOPs of RECLIP with the baseline method in Table 2. The baseline method, CLIP, our repro., requires 71.4 GFLOPs. Our RECLIP-80 reduces GFLOPs by $\sim$ **7×** to **10.1** and **RECLIP-64** further reduces GFLOPs by $\sim$ **10×** by using even smaller images.

## 4.2 System-level Comparison

We present system-level comparison between RECLIP and a series of existing methods on Flickr30K and MSCOCO image-text retrieval benchmarks, and ImageNet classification accuracy in Table 3. We train RECLIP for 600k steps and then finetune for 50k steps with the image size of $448 \times 448$. For RECLIP-64-F20K, we finetune for 20k steps.

Table 3: Comparisons of zero-shot image-text retrieval and ImageNet classification top-1 accuracy on Flickr30K, MSCOCO and ImageNet. Models that use the fully-supervised dataset (Sun et al., 2017) and much larger are marked in gray. †: We refer to (Li et al., 2022b) to convert GPU cost to TPU usage in CLIP (Radford et al., 2021), FILIP (Yao et al., 2021). Cores×hours results are reported on TPU-v3 infrastructure. Best results are **bolded**.

| Method | Image Encoder Size | Cores × Hours | ImageNet Top-1 | Flickr30K (1K test set) | | | | MSCOCO (5K test set) | | | |
| | | | | image-to-text | | text-to-image | | image-to-text | | text-to-image | |
| | | | | R@1 | R@5 | R@1 | R@5 | R@1 | R@5 | R@1 | R@5 |
| PaLI (Chen et al., 2022b) | 3.9B | 598.7K | 85.4 | - | - | - | - | - | - | - | - |
| BASIC (Pham et al., 2021) | 2.4B | 288.1K | 85.7 | - | - | - | - | - | - | - | - |
| CoCa (Yu et al., 2022) | 1B | 962.1K | **86.3** | 92.5 | 99.5 | 80.4 | 95.7 | 66.3 | 86.2 | 51.2 | 74.2 |
| CLIP (Radford et al., 2021) | 302M | 120.0K† | 76.2 | 88.0 | 98.7 | 68.7 | 90.6 | 58.4 | 81.5 | 37.8 | 62.4 |
| ALIGN (Jia et al., 2021) | 408M | 355.0K | 76.4 | 88.6 | 98.7 | 75.7 | 93.8 | 58.6 | 83.0 | 45.6 | 69.8 |
| FILIP (Yao et al., 2021) | 302M | 180.0K† | **78.3** | 89.8 | **99.2** | 75.0 | 93.4 | 61.3 | 84.3 | 45.9 | 70.6 |
| FLIP (Li et al., 2022b) | 303M | 81.9K | 75.8 | 91.7 | - | 78.2 | - | 63.8 | - | 47.3 | - |
| **RECLIP-80 (ours)** | 303M | 28.7K | 76.3 | 91.4 | 99.1 | **79.2** | 94.7 | **64.9** | **85.2** | **48.2** | **72.6** |
| **RECLIP-64-F20K (ours)** | 303M | **16.4K** | 75.3 | **92.5** | 99.1 | 78.7 | **94.9** | 64.5 | **85.2** | 47.3 | 71.9 |

Table 4: LVIS open-vocabulary object detection. RECLIP maintains the same open-vocabulary detection (AP$_r$) and standard detection (AP) as the state of the art RO-ViT despite using much less training resources.

| ViT based method | Pretrained model | Detector backbone | AP$_r$ | AP |
| --- | --- | --- | --- | --- |
| RO-ViT (Kim et al., 2023) | ViT-L/16 | ViT-L/16 | **32.1** | 34.0 |
| **RECLIP-RO-ViT (Ours)** | ViT-L/16 | ViT-L/16 | 32.0 | **34.7** |

From Table 3, we observe clear resource savings and highly competitive performance achieved with our simple and efficient training recipes. RECLIP with small images saves $3 \sim 59\times$ compute resource in cores×hours. When comparing to the models with the similar scale of the image encoder (Radford et al., 2021; Jia et al., 2021; Yao et al., 2021; Li et al., 2022b), RECLIP reduces resource use by $5 \sim 22$ times with competitive zero-shot retrieval and image classification performance. In the comparisons to FLIP (Li et al., 2022b), RECLIP-64-F20K uses $\sim 5\times$ less resource in cores×hours and outperforms it by +2.0 on Flickr30k and MSCOCO retrieval. Surprisingly, when compared to the CoCa, RECLIP-64-F20K significantly saves $\sim 98\%$ resource use and achieves the best image to text retrieval on Flickr30K test set, giving 92.5 of R@1. RECLIP-64-F20K gives 75.3, which is very competitive on zero-shot ImageNet classification among purely language supervised approaches. We believe this resource savings mostly come from the use of very short image sequence length i.e., 16, which is very different from existing recipes (Radford et al., 2021; Li et al., 2022b; Yu et al., 2022; Zhai et al., 2022).

We also observe that RECLIP-80 uses $3 \sim 34\times$ less compute resource. When comparing to the CoCa (Yu et al., 2022), RECLIP-80 saves $\sim 97\%$ resource use and achieves highly competitive retrieval performance. The resource savings of RECLIP-80 can also be attributed to the largely-reduced sequence length, i.e., **25** for the image encoding. RECLIP-80 achieves highly competitive ImageNet top1 accuracy of 76.3, which outperforms CLIP and is on-par with ALIGN. Overall, RECLIP provides very affordable recipes for large-scale language and image pretraining.

We note that some leading methods (Chen et al., 2022b; Pham et al., 2021; Yu et al., 2022) marked in gray demonstrate substantially better zero-shot classification because of larger image encoder capacity and the use of JFT (Sun et al., 2017) dataset. JFT is a human-annotated classification dataset which is cleaner than most web crawled image-text datasets (Radford et al., 2021; Schuhmann et al., 2021; Jia et al., 2021) and most advantageous for zero-shot classification, so we list the JFT-trained entries there for reference only.

## 4.3 Open Vocabulary Detection

We conduct evaluation on the LVIS dataset (Gupta et al., 2019) by using RECLIP for open vocabulary detection. We take a recent SOTA approach RO-ViT (Kim et al., 2023) as the baseline and apply RECLIP-80 to pre-train the model (RECLIP-RO-ViT). We train only on the LVIS base categories (frequent & common) and test on both the base and novel (rare) categories following the protocol of ViLD (Gu et al., 2022). The results are in the Table 4. RECLIP-RO-ViT achieves 32.0 Mask APr (AP on rare categories) (Gupta et al., 2019), matching the state of the art performance of RO-ViT (32.1). This is surprisingly encouraging because detection task typically requires much higher resolution e.g. 1024 than classification task to recognize the small objects, which can be especially challenging for RECLIP due to the low-res information loss. In addition, RECLIP-RO-ViT outperforms RO-ViT by 0.7 on all-category AP, showing

that its representation is also suitable for standard detection on the base categories. These detection results suggest that RECLIP representation is versatile and suitable for a broader range of object and pixel-level tasks.

## 4.4 Ablations

In this section, we ablate the design of RECLIP training and evaluate on the zero-shot retrieval and classification accuracy.

**The importance of high-resolution finetuning.** Table 5 shows the importance of finetuning RECLIP with high-resolution data after the main training phase. We compare the retrieval and classification accuracy by using the model trained with and without high-resolution finetuning on an image size of 224 for 50k steps. We observe that high-resolution finetuning significantly improves the performance for zero-shot retrieval and classification. In particular, training RECLIP by using the smallest images, e.g. $64 \times 64$, high-resolution finetuning offers the most notable benefits. This is also aligned with the results in Table 3 where RECLIP models trained with small images, e.g. $64 \times 64$ or $80 \times 80$, and finetuned with $448 \times 448$ for a short cycle can achieve comparable performance with SOTA models.

Table 5: The importance of RECLIP high-resolution finetuning. We found that high-resolution finetuning significantly improves zero-shot transfer performance. RECLIP-X: RECLIP trained with image size $X$. Best results are **bolded.**

| | Total Training Steps | Before high-resolution finetuning | | | | | After high-resolution finetuning | | | | |
| | | INet Top-1 | Flickr30K | | MSCOCO | | INet Top-1 | Flickr30K | | MSCOCO | |
| | | | I2T | T2I | I2T | T2I | | I2T | T2I | I2T | T2I |
|---|---|---|---|---|---|---|---|---|---|---|---|
| RECLIP-112 | 300k | **69.0** | **83.2** | **67.9** | **58.6** | **40.0** | 74.2 (+5.2) | 90.0 (+6.8) | 76.6 (+8.7) | **63.1 (+4.7)** | 45.0 (+5.0) |
| RECLIP-80 | 300k | 66.3 | 80.8 | 65.4 | 54.6 | 37.4 | **74.3 (+8.0)** | **91.0 (+10.2)** | **77.1 (+11.7)** | 62.8 (+8.2) | **45.7 (+8.3)** |
| RECLIP-64 | 300k | 62.8 | 79.6 | 63.6 | 51.4 | 34.5 | 73.3 (+10.5) | 89.4 (+9.8) | 77.0 (+6.4) | 62.2 (+10.8) | 45.2 (+10.7) |
| RECLIP-112 | 600k | **70.7** | **87.4** | **71.9** | **59.0** | **40.9** | **75.8 (+5.1)** | 90.6 (+3.2) | 77.6 (+5.7) | 63.6 (+4.6) | 46.5 (+5.5) |
| RECLIP-80 | 600k | 67.7 | 82.8 | 68.1 | 55.8 | 39.0 | **75.8 (+8.1)** | **91.3 (+8.3)** | **78.2 (+10.1)** | **64.6 (+ 8.8)** | **47.2 (+8.2)** |
| RECLIP-64 | 600k | 65.5 | 80.9 | 66.1 | 54.3 | 37.1 | 75.4 (+9.9) | 91.0 (+10.1) | 78.1 (+12.0) | 64.2 (+10.1) | 46.9 (+9.8) |

**Text length for RECLIP main training.** Table 6 studies the text length for the RECLIP training. We use the text length of 64 and 16 to train our RECLIP with an image size of 80. Somewhat surprisingly, we observe that using a short text length, i.e. 16, during the main training phase clearly reduces the resource use and achieve competitive zero-shot retrieval and image classification performance. This training efficiency gains is possible because we use much shorter image sequence lengths than existing recipes (Radford et al., 2021; Yu et al., 2022).

Table 6: The effect of the text length in RECLIP main training. We found that using a short image sequence can further save compute resource and achieve promising zero-shot transfer performance. Default RECLIP settings are in dark gray . Best results are **bolded**.

| Text Length | Cores × hours | Flickr30K | | MSCOCO | | INet Top-1 |
| | | I2T | T2I | I2T | T2I | |
|---|---|---|---|---|---|---|
| 64 | 15.5K | 91.2 | 78.0 | 64.3 | 46.7 | 75.6 |
| 16 | **11.2K** | **91.3** | **78.2** | **64.6** | **47.2** | **75.8** |

**Small batch size for RECLIP main training phase.** Our RECLIP is designed with principles of using constant batch size but varying image resolutions during the main training phase. Table 7 ablates effects of the batch size during the main training phase on zero-shot retrieval and image classification accuracy. We first train the model for 250k steps by using the batch size of 4k or 16k and the image size 112; then we finetune it for 50k steps by using the batch size of 16k and the image size of 224. From Table 7 shows that using smaller batch size (4k) saves compute resource by 69%, but the zero-shot retrieval and classification performance drops significantly even with the same high-resolution finetuning phase. Therefore, we conclude that using the same large batch size is important for language image pretraining to ensure competitive zero-shot transfer performance.

Table 7: The importance of RECLIP main training with constant batch size. We found that using the same batch size (16k) for RECLIP main training and finetuning achieves better zero-shot transfer performance. Default RECLIP settings are in dark gray . Best results are **bolded**.

| Batch Size | Cores × Hours | Flickr30K | | MSCOCO | | INet Top-1 |
| | | I2T | T2I | I2T | T2I | |
|---|---|---|---|---|---|---|
| 4k | **4.2K** | 81.9 | 68.8 | 51.2 | 38.6 | 64.4 |
| 16k | 13.1K | **90.0** | **76.6** | **63.1** | **45.0** | **74.2** |

**Increasing the batch size with small images for RECLIP**. In Table 8, we ablate RECLIP by varying both the batch size and image size during the main training phase. The multi-grid training paradigm is as below: (1) we equally divide training process into 3 stages with the same steps in each; (2) we train the model for 25k, 50k and 100k steps by using the batch size of 64k, 32k and 16k, and the image size of 112, 160 and 224 in each stage. The idea is to increase the batch size while using low resolution data, and decrease the batch size with high-resolution data. The multi-grid free baseline is trained for 300k steps by using a constant batch size 16k and image size 112, and finetuned with image size 224. We observe that RECLIP without "MG" is not only simpler, but saves computational resource by 30%. In addition, RECLIP achieves better zero-shot retrieval retrieval performance on Flickr30K and MSCOCO and very similar ImageNet performance.

Table 8: The effect of multigrid training strategy, where we increase the image size and decrease the batch size simultaneously. We found RECLIP is simple and effective. Default RECLIP settings are in dark gray .

| MG | Cores× Hours | Flickr30K | | MSCOCO | | INet |
| | | I2T | T2I | I2T | T2I | Top-1 |
| --- | --- | --- | --- | --- | --- | --- |
| ✓ | 18.4K | 89.2 | 75.5 | 62.3 | **45.3** | **74.5** |
| ✗ | **13.1K** | **90.0** | **76.6** | **63.1** | 45.0 | 74.2 |

**Multi-stages RECLIP high-resolution finetuning.** In Table 9, we further study RECLIP with 1 and 2 high-resolution finetuning stages given a model trained with low-resolution data. We study the following two variants. $(112 \rightarrow 224 \rightarrow 448)$: we train the model for 300k steps with the image size of 112, finetune it for 40k steps with the image size of 224, and finetune it for another 40k steps with the image size of 448. $(112 \rightarrow 448)$: we train the model for 300k steps with the image size of 112 and finetune it for 50k steps with the image size of 448. We set 50k steps to keep the computation cost comparable with the first one. We observe that $(112 \rightarrow 448)$ gives very competitive zero-shot retrieval and image classification accuracy. Thus, we use only one high-resolution finetuning stage.

Table 9: RECLIP with one-stage or multi-stages high-resolution finetuning. We found that one high-resolution finetuning stage is simple and sufficient. Default RECLIP settings are in dark gray . The best results are **bolded**.

| Stages | Core× Hours | Flickr30K | | MSCOCO | | INet |
| | | I2T | T2I | I2T | T2I | Top-1 |
| --- | --- | --- | --- | --- | --- | --- |
| $112 \rightarrow 224 \rightarrow 448$ | 31.1K | 91.0 | 77.7 | 64.1 | **47.4** | **76.2** |
| $112 \rightarrow 448$ | **30.8K** | **90.7** | **78.0** | **64.3** | 47.0 | 76.1 |

**Comparisons of image resizing and token masking** In Table 10, we present a comparison between token masking (Li et al., 2022b) and image resizing training strategy with matching computational budget. The benchmark is zero-shot ImageNet classification. All factors other than masking vs resizing are controlled to be the same. For example, we use the same batch size, data, training recipe, and the same number of iterations for low-resolution (vs masked) pretraining and high-resolution (vs unmasked) finetuning. To match the compute usage betweeen resizing and masking, we set the masking ratios such that the sequence lengths are the same. For example, Mask-112 masks 75% tokens to match the sequence length of RECLIP-112 (assuming the baseline using full image size 224x224). Table 10 shows that image resizing has a clear advantage over token masking. RECLIP-112 starts with a gap of +2.9 with Mask-112. As the token masking ratio goes above 75% (Mask-112), we observe an increasing gap between resizing and masking (+5.4% for RECLIP-64), showing the clear advantage of resizing in very low-compute settings.

Table 10: Comparison of resizing vs token masking on zero-shot ImageNet classification. RECLIP-X: RECLIP with image size X. Mask-X: token masking with the same compute budget as the corresponding RECLIP-X. Resizing consistently outperforms masking, and the gap increases with decreasing compute budget. Best results are **bolded**.

| X (Image Size) | Mask-X | RECLIP-X |
| --- | --- | --- |
| 112 | 72.9 | **75.8 (+2.9)** |
| 80 | 71.3 | **75.8 (+3.5)** |
| 64 | 69.5 | **74.9 (+5.4)** |

## 4.5 Visualization

**Visualization of small images.** In Fig. 3, we visualize images at various resolutions paired with their corresponding texts. We observe that small images generally preserve high-level structures of the original images, and contain

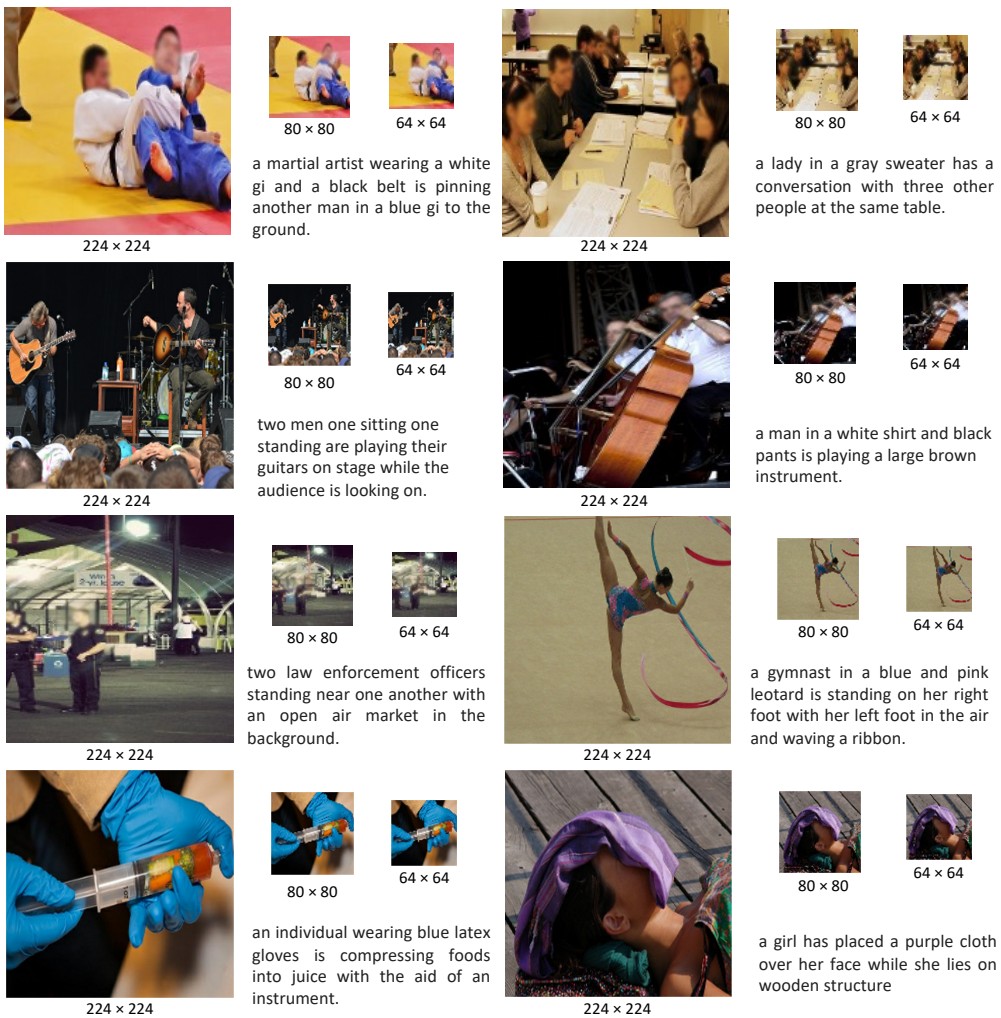

Figure 3: Visualization of image-text pairs and images are in various resolutions. Images are scaled with the same factor of 0.01 for both height and width. Small images contain sufficient visual information for contrastive training.

sufficient visual information for language supervisions. For example, the martial arts, office meeting, concert, and gymnastics scenes are clearly recognizable down to $64 \times 64$ resolution. This supports the key insight of our RECLIP training design that leverages small images for the main training phase to save computation.

**Visualization of image and text retrieval.** We present image and text retrieval results of RECLIP in Fig. 4. Despite highly resource efficient training, RECLIP still produces accurate results on both image-to-text and text-to-image retrieval. For example, the concepts of football players, race cars, circular sculpture, police officer, musicians, and bulldozer are all correctly matched between image and texts.

## 5 Conclusions

We present the RECLIP, a method for resource-efficient language image pretraining. We propose to leverage small images with paired texts for the main constrastive training phase and finetune the model with high-resolution images for a short cycle at the end. The proposed training method has been validated on zero-shot image and text retrieval benchmarks and image classification datasets. In comparisons to the baseline method, RECLIP training recipe saves the computations by $6 \sim 8\times$ with improved zero-shot retrieval performance and competitive classification accuracy. Compared to the state-of-the art methods, RECLIP significantly saves **79**% $\sim$ **98**% resource in cores×hours with

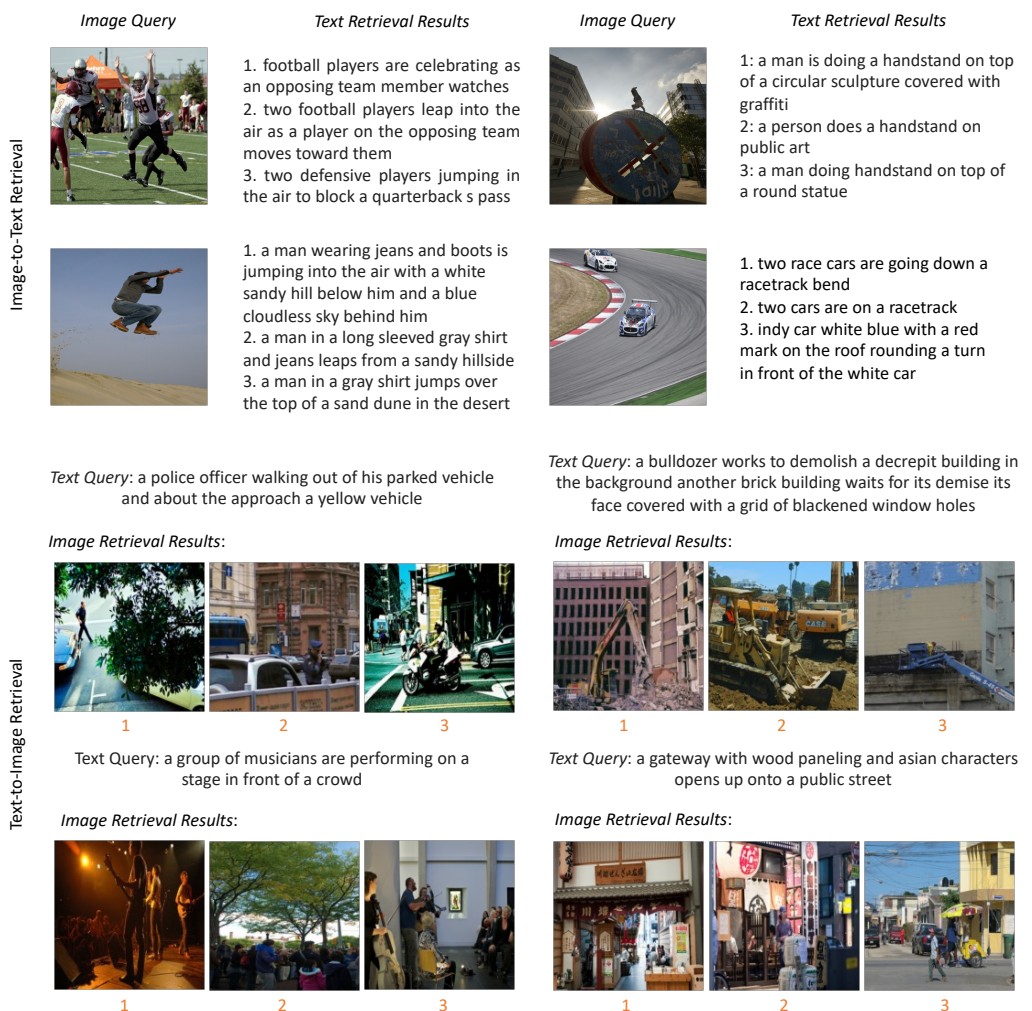

Figure 4: Visualization of image and text retrieval results. Despite training with orders of magnitude less resource, RECLIP correctly match many visual concepts with texts.

highly competitive zero-shot classification and image-text retrieval performance. We hope RECLIP paves the path to make contrastive language image pretraining more resource-friendly and accessible to the broad research community.

## Broader Impact Statement

Language image pretraining plays an important role in many applications, e.g. image and text retrieval, text-to-image generations, open-vocabulary detection, etc. This work presents a language image pretraining method, RECLIP, on large-scale web datasets and the proposed model has been evaluated on a series of zero-shot downstream tasks. The large image-text corpus may contain biased or harmful content which could be learnt by the model. Our model is for research use only and these models should not be used in applications that involve detecting features related to humans (e.g. facial recognition). The good news is RECLIP significantly reduces the resource use, thereby reducing the carbon footprint and is very environment-friendly for the community to build upon in the long run.

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
