# OpenReview forum: "RECLIP: Resource-efficient CLIP by Training with Small Images"
_TMLR — Accepted by TMLR_

### Review · Reviewer_SQx8 · 2023-04-27

**Summary Of Contributions:**

This paper proposes to split CLIP pre-training up into two parts, the first of which uses very small (64x64) images leading to incredible resource savings. The second part is high-res fine-tuning, which resembles standard pre-training. Overall this paper takes a similar approach as FLIP (https://arxiv.org/abs/2212.00794) but instead of randomly masking patches it makes the images smaller.

**Audience:**

Yes

**Claims And Evidence:**

Yes

**Requested Changes:**

It would further strengthen the paper to see an apples-to-apples comparison with FLIP, i.e., how does masking compare to resizing with the same computational budget, across computational budgets?

-- Minor typo like changes --
In "Implementation details" in Sec 3, the authors say learning rate 1 x e^{-4} but I believe they mean 1e-4 which is 10^{-4} written in scientific notation.

**Strengths And Weaknesses:**

Strengths:
- The approach is intuitive, straightforward to implement, and I believe will be very useful to the community
- Great resource savings over baseline
- Better retrieval performance
- Many ablation studies

Weaknesses:
- No apples-to-apples comparison to FLIP

---

> ### Author Response · Authors · 2023-06-13
> **Author feedback for Reviewer SQx8**
>
> Thank you very much for the helpful review. Please find our response as below.
>
> >**Comparison of image resizing (RECLIP) vs token masking (FLIP):**
>
> We appreciate the reviewer’s suggestion and have conducted an apple-to-apple comparison between masking and resizing contrastive learning strategy with matching computational budget. All factors other than masking vs resizing are controlled to be the same. For example, we use the same batch size, data, training recipe, and the same number of iterations for low-resolution (vs masked) pretraining and high-resolution (vs unmasked) finetuning. To match the compute usage betweeen resizing and masking, we set the masking ratios such that the sequence lengths are the same. For example, Mask-112 masks 75% tokens to match the sequence length of RECLIP-112 (assuming the baseline using full image size 224x224).
>
> The Table below shows that image resizing has a clear advantage over token masking. RECLIP-112 starts with a gap of +2.9 with Mask-112. As the token masking ratio goes above 75% (Mask-112), we observe an increasing gap between resizing and masking (+5.4% for RECLIP-64), showing the clear advantage of resizing in very low-compute settings.
> | X | Mask-X | RECLIP-X |
> | ------- | -------- | -------- |
> | 112 | 72.9 | 75.8 (+**2.9**)|
> | 80 | 71.3 | 75.8 (+**3.5**) |
> | 64 | 69.5 | 74.9 (+**5.4**) |
>
> We have included this ablation in the revised manuscript.
>
> >**Minor typos:**
>
> We thank the reviewer for catching these typos and have corrected them in the revised manuscript.

---

> > ### Comment · Reviewer_SQx8 · 2023-06-28
> > **Thank you for the ablation.**
> >
> > My concerns are addressed.

---

### Review · Reviewer_tV4K · 2023-05-02

**Summary Of Contributions:**

Summary:

This paper presents a simple but effective framework for efficient CLIP model training. Upon the assumption that low-resolution images have sufficient information about their content description, training the CLIP model in the size of 64x64 or 80x80 and token length 16 would greatly reduce the computation cost (by 6 to 8x less). Moreover, it has surprisingly achieved comparable or even better results compared with the baseline models trained by high-resolution data. Such a low-resolution-based model can be easily boosted by simple finetuning over high-resolution data. In all, the proposed RECLIP model achieves a good balance between computation and performance.



**Audience:**

Yes

**Claims And Evidence:**

Yes

**Requested Changes:**

- For Tab. 1, 3, and 4, please highlight the best results in the table.

- For Tab. 4, please add the columns of performance improvements (+X.X) after high-resolution finetuning.

- Even though the proposed method has achieved good results on Flickr30K, MSCOCO, and ImageNet, some more challenging benchmarks, such as Winoground [1], are encouraged.

[1] Thrush, Tristan, et al. "Winoground: Probing vision and language models for visio-linguistic compositionality." Proceedings of the IEEE/CVF Conference on Computer Vision and Pattern Recognition. 2022.

**Strengths And Weaknesses:**

Pros:
+ This proposed method is simple and effective. It would raise the high interest in training models on low-resolution data in the community.

+ CLIP model is a foundation model that has a very large impact on both industry and academia. This paper extends CLIP with non-trivial design and findings.

+ This paper is well-written with clear logic. The discussion of related works is comprehensive.

+ This paper has solid and detailed experimental results and analysis. The comparison with baselines is fair and inspiring.

Cons/Questions:

- The proposed method is super simple, and it looks like an engineering trick even though it is effective and useful. More insightful discussions or theoretical analyses are welcome to enrich the content.

- Despite that this paper takes the same model architecture as previous papers, a brief description of the neural network is welcome as some readers may not read previous papers.

- For high-resolution image finetuning, how many images are needed? Are the high-resolution the same content as low-resolution images?

- Since this paper used TPU-v3 for model training, it would be better to explain the capacity of TPU vs GPU since most of the readers are not familiar with TPU.

---

> ### Author Response · Authors · 2023-06-13
> **Author feedback for Reviewer tV4K**
>
> Thank you very much for the helpful review. Please find our response as below.
>
> >**Theoretical analysis of the proposed method:**
>
> Thank you for the suggestion to enrich the content. Our method is inspired by the idea of coarse-to-fine in computer vision literature. We leverage small images to efficiently learn large-scale image-language correspondence, and then finetune the model with high-resolution data to help it learn the important details. We believe the effectiveness and computation savings of RECLIP can be valuable to the community.
>
> We expanded the theoretical analysis on the complexity of RECLIP in the revised manuscript. The existing complexity analysis is based on the self-attention operations in transformers. However, empirically the complexity of a transformer may not be dominated by the self-attention layers, because the feed forward layers also play an important role. GPT-3 [1] paper have provided computation analysis of their language models, where the computation cost is assumed to scale as $O(N)$, linear with the sequence length $N$ due to the feed forward layers. Thus, here we discussed the lower-bound of the complexity $C_{lb}$ of RECLIP following the same assumption of GPT-3 [1]. Using the notation of Eq. 3 and 4 of the revised manuscript, we have
> \begin{equation}
> C_{lb} = O(BN) = O(\frac{BHW}{pr^2})
> \end{equation}
> During the training, the B, H, W, p are normally constant, so the above equation can be simplified as:
> \begin{equation}
> C = O(\frac{1}{r^2})
> \end{equation}
> Compared to Eq. 5, we observe that the computation savings in practice may be somewhere between $O(\frac{1}{r^2})$ and $O(\frac{1}{r^4})$. This suggests that changing $r$ is still very effective for compute savings. We have added the new analysis accordingly in *Training complexity* section of Section 3.2.
>
> >**Network architecture details:**
>
> We use the ViT-Large backbone as image encoder by default. The ViT-Large is a vision transformer which consists of 24 multi-head self-attention layers with 16 heads and the width dimension of 1024. We have added descriptions of the vision transformers and the text tower architecture in *Network architecture* section of Section 3.2.
>
> >**Number of images for high-resolution finetuning:**
>
> The number of images for high-resolution finetuning is ~800 million. The total number of images for low-resolution training and high-resolution finetuning combined is the same as in the baseline method. The images used for high-resolution training are the same as those for the low-res training, except that we remove the downsampling to preserve the rich visual details. We have added clarification in *high-resolution finetuning* section of Section 3.2.
>
> >**TPU vs. GPU for training:**
>
> Thanks for the question. [2] provides a good overview of TPU vs GPU comparison. We summarize some key points below. GPU is a general-purpose processor to support many applications and software. For every calculation a GPU must access register or shared memory to read operands and store the intermediate calculation results. In contrast, TPUs are specialized for neural network workloads and handle massive matrix operations used in neural networks at fast speeds. According to the report [3], TPU v3 achieves high-performance training across multiple workloads. For example, training the ResNet-50 for classification, using 16 TPU v3 chips for training and another 4 TPU v2 chips for evaluation is 19% faster than using a DGX-2 machine with 16 V100 GPUs. Each TPU v3 device has 16GB high-bandwidth memory per core, which is comparable to that of a V100 and suitable for synchronous large-scale training. We have included more clarification in *implementation details* of Section 3.2.
>
> >**Highlight best results:**
>
> We appreciate the reviewer’s suggestion. We have highlighted the best results for all tables in the revised manuscript.
>
> >**Add performance improvement:**
>
> We appreciate the reviewer’s suggestion. We have added the performance improvements in Table 5 (old Table 4) in the revised manuscript.
>
> >**Evaluation with fine-grained dataset Winoground:**
>
> We appreciate the reviewer’s suggestion. We agree that Winoground is a valuable benchmarkfor fine-grained compositional reasoning and it would be interesting to compare our method with CLIP on it. Unless the use of low-res pre-training has a surprising effect on compositional reasoning, we think the comparison with CLIP on Winoground may show similar trend as the comparison on ImageNet and Flickr/COCO. We will do our best to include an evaluation on Winoground in the final version.
>
> >**References**
>
> [1]. Brown et al. Language Models are Few-Shot Learners. NeurIPS, 2020
> [2]. https://cloud.google.com/tpu/docs/intro-to-tpu
> [3]. https://cloud.google.com/blog/products/ai-machine-learning/mlperf-benchmark-establishes-that-google-cloud-offers-the-most-accessible-scale-for-machine-learning-training

---

### Review · Reviewer_pEXU · 2023-05-29

**Summary Of Contributions:**

The authors perform an empirical study to train current image-text contrastive models like CLIP with fewer computational resources.
The proposed method, named "Resource Efficient Contrastive Language Image Pre-training" (RECLIP),
trains using small resolution images $64 \times 64$ for most of the training duration and fine-tuning with higher resolution in the end.
This design choice is different than existing CLIP-style models that typically train with $224 \times 224$ resolution.
The authors motivate this by a simple observation: for an $r \times$ reduction in input image resolution,
the forward pass through a vision transformer has complexity $O(1/r^4)$.

The authors present a series of controlled comparisons between CLIP and RECLIP models at different image resolutions,
and evaluate them on standard vision tasks like zero-shot image classification and retrieval.
RECLIP is shown to match the baseline performance despite using significantly fewer computational resources.

**Audience:**

Yes

**Broader Impact Concerns:**

The authors have included a brief statement on the broader impact of their work.
In my opinion, their address is overall sufficient.
I agree with the authors that this work can have a positive impact on reducing the carbon footprint
in training large vision-language models that are becoming increasingly popular in recent times.

All models are trained on a private dataset comprising millions of image-text pairs —
such data may contain harmful stereotypes targeted to specific population subgroups based on their gender and racial identity.
However, the main goal of this goal is to observe the resource savings of their approach
as compared to baselines trained using the exact same data.
Putting this in context, the authors state that their models must be used for research only.
I would suggest that the authors make the wording more precise —
these models should now be used in applications that involve detecting features related to humans (e.g. facial recognition).


**Claims And Evidence:**

Yes

**Requested Changes:**

I have ordered my concerns in approximate order of how important they are to the core contributions of the paper.
I request the authors to address them as per their capacity and available computational resources.
Weakness (1) is crucial from a practical standpoint, whereas weaknesses (2 and 3) could further improve the paper.

Typo on Page 2: "iamge" -> "image"

**Strengths And Weaknesses:**

### Strengths

This paper has a number of notable strengths; below I list a few salient strengths with supporting evidence:

1. **Simplicity:**
The proposed method is very simple and intuitive.
Practically speaking, any existing CLIP implementation can be repurposed to RECLIP with minimal modifications.
Moreover, the proposed method can be additive to a broad set of CLIP extensions in published literature
like SLIP (Mu et al, 2022), FILIP (Yao et al, 2022), MERU (Desai et al, 2023), etc.
These contributions can be impactful in the long run, regardless of which underlying CLIP method stands the test of time.

2. **Reasonable empirical results:**
RECLIP shows a reasonable trade-off between training resources (and speed) vs. performance on many diverse downstream vision tasks.
I recognize that RECLIP _cannot_ be expected to outperform CLIP in absolute terms under controlled comparisons,
since it is observed strictly less information than CLIP (lossy compression of input images).
Main experiments and all ablations are adequate and thorough.

3. **Eliminates CLS token to reduce overhead:**
The authors switch from the ubiquitous design of ViTs that use a CLS token to aggregate global image features
and instead prefer using global average pooling. I believe that this choice is co-designed with using TPU-v3
to conduct experiments — prior work has noted almost 50% extra overhead due to the CLS token (https://arxiv.org/abs/2106.04560).
I believe this is a clever choice, but not entirely obvious to every reader —
the authors should consider including this discussion and cite prior works that also make this observation.

4. **Clarity in writing and presentation:**
The training setup and modeling components are introduced with clear exposition.
All figures are neat and minimal (although can be made more compact),
and result tables have one takeaway message each.
The authors have also extensively covered the recent related work in this area.
Overall, I think the writing clarity and presentation are excellent.
I hope that the authors open-source the code and models upon acceptance.

### Weaknesses

1. **Discrepancy between theoretical and hardware-specific cost savings of RECLIP:**
This concern is related to strength (3).
Section 3.2 of "Scaling ViTs" paper (https://arxiv.org/abs/2106.04560) states that TPU-v3
pads tensors to nearest multiples of 128.
Hence, using CLS token with ViT-L/14 (1 + 14x14 = 257 visual tokens) causes 50% overhead due to padding.
I wonder if this padding feature in TPU-v3 also affects RECLIP.
For instance, RECLIP-64 has 16 only visual tokens, are they padded to 128 tokens internally,
and hence RECLIP-64 and RECLIP-80 (16 and 25 tokens respectively) have approximately the same duration of forward pass?
Please correct me if I am wrong.

2. **Why does RECLIP-112 perform worse than RECLIP-80?**
Results in Tables 1 and 4 suggest this for multiple downstream tasks.
Intuitively, one would expect a steady improvement in performance with higher image resolution
due to less loss in image compression. Do the authors have an intuition for this?

3. **Evaluating RECLIP on object detection:**
I believe this paper lacks a crucial evaluation, which could further make it stronger — object detection.
Modern object detectors using ViT backbones operate on very high image resolution
(e.g. ViTDet uses $1024 \times 1024$, https://arxiv.org/abs/2203.16527).
Since RECLIP flexibly allows different image resolutions for training and fine-tuning,
can RECLIP backbones show better transfer on object detection (classical and open-vocabulary)
when fine-tuned with higher resolutions?

4. **Why learn the positional embeddings? (minor suggestion)**
RECLIP trains the image encoder at two different resolutions.
When transitioning to higher resolution, the position embeddings are upsampled and fine-tuned.
I wonder if the authors considered using a fixed sine-cosine position embedding,
which is a naturally smooth function that can be kept fixed and scaled to any resolution without aliasing effects.

5. **Figure 1 and 2 ordering (minor suggestion):**
These figures may be interchanged for better readability.
It is important to expose the central contribution to the reader in a figure before the results.
Figure 2 is also referenced in the Introduction text before Figure 1.
Feel free to ignore this suggestion.

---

> ### Author Response · Authors · 2023-06-13
> **Author feedback for Reviewer pEXU**
>
> Thank you very much for the helpful review. Please find our response as below.
>
> >**Effects of TPU padding:**
>
> Great points about CLS token and TPU padding. We follow the [performance guidance on cloud TPU](https://cloud.google.com/tpu/docs/performance-guide#padding) and make sure the feature dimension is a multiple of 128 (i.e. 4096) and the batch size is a multiple of 1024 (i.e. 16384), so both conditions are met to avoid padding on sequence dimension and minimize overhead. In practice, we observe a speedup of ~40% (relative improvement) by using RECLIP-64 as opposed to RECLIP-80 during the first low-res training stage, which suggests that the shorter sequence of RECLIP-64 is beneficial. We have added clarification about this in the implementation details.
>
> >**Empirical observations of RECLIP-112:**
>
> Thanks for the observation. We agree that higher-resolution images should lead to improved results in general. The results of our low-res training stage matches the intuition, where RECLIP-112 > RECLIP-80 > RECLIP-64 in all metrics (see Table 5).
> However, we observe all RECLIP variants achieve very similar retrieval/classification performance after high-res finetuning, even though the ranking among the variants does not perfectly match the ordering of image size used (see Table 5). We think this is partly due to the effectiveness of high-res finetuning to recover the representation quality loss of low-res training, which is a key finding of this paper. At this moment, we do not have concrete explanations for the mixed ordering after high-res finetuning. It is an empirical question whether a different set of hyper-parameters, data, and infrastructure may yield the expected ordering, although we expect the effectiveness of high-res finetuning to remain similar. We will do our best to gain a better understanding of this behavior for the final version.
>
> >**RECLIP for open-vocabulary detection:**
>
> Thanks for the suggestion. We conduct evaluation on the LVIS dataset [1] by using RECLIP for open vocabulary detection. We take a recent SOTA approach  RO-ViT [2] as the baseline and apply RECLIP-80 to pre-train the model (RECLIP-RO-ViT). We train only on the LVIS base categories (frequent & common) and test on both the base and novel (rare) categories following the protocol of ViLD [3]. The results are in the table below. RECLIP-RO-ViT achieves 32.0 Mask APr (AP on rare categories) [1] and matches state of the art RO-ViT (32.1). This is surprising and encouraging because detection task typically requires even higher resolution e.g. 1024 than classification task to recognize the small objects, which can be especially challenging for RECLIP due to the low-res information loss. In addition, RECLIP-RO-ViT outperforms RO-ViT by 0.7 on all-category AP, showing that its representation is also suitable for standard detection on the base categories. These detection results suggest that RECLIP representation is versatile for a broader range of object/pixel-level tasks.  We have updated the abstract, introduction, and experiments to highlight these results.
>
> | Vit based method | Pretrained model | Detector backbone | APr | AP |
> | ---------------- | ---------------- | ---------------- | --- | ---|
> | RO-ViT [2] | ViT-L/16 | ViT-L/16 | 32.1 | 34.0 |
> | RECLIP-RO-ViT (Ours) | ViT-L/16 | ViT-L/16 | 32.0 | 34.7 |
>
>
> >**RECLIP with different positional embeddings:**
>
> Thank you for the suggestion on exploring different positional embedding (PE) techniques with RECLIP. We choose learnable positional embeddings following the standard practice with vision transformers [Dosovitskiy et al 2021] and CLIP [Radford et al 2021]. We have compared trainable vs fixed SinCos PE on image-text retrieval using our own CLIP training recipes. Our results show that the trainable version is 0.6 point better on average across i2t/t2i Recall@1 metrics on Flickr/COCO. We agree it is an interesting aspect to understand how different PEs compare to learnable PE in RECLIP, and will do our best to include this in the final version. We have added clarification about positional embeddings in the high-resolution finetuning section.
>
> >**The order of Figure 1 and 2:**
>
> Thank you for the feedback. We have re-organized the order of Figure 1 and Figure 2 in the revised manuscript.
>
> >**Typos on page 2:**
>
> Thank you. We have corrected these errors in the revised manuscript.
>
> >**Broader Impact Concerns on usage purpose of models:**
>
> We fully agree and have updated the description on usage purpose of models in the section of *Broader Impact Concerns*.
>
> >**References:**
>
> [1]. Gupta et al. LVIS: A Dataset for Large Vocabulary Instance Segmentation. CVPR 2019.
> [2]. Kim et al. Region-Aware Pretraining for Open-Vocabulary Object Detection with Vision Transformers. CVPR 2023.
> [3] Gu et al. Open-Vocabulary Open Detection via Vision and Language Knowledge Distillation. ICLR 2022.

---

> > ### Comment · Reviewer_pEXU · 2023-06-26
> > **Thank you for the response!**
> >
> > I have read the rebuttal and I find all the authors' responses convincing. I thank the authors for clarifications and additional experiments. I recommend acceptance.

---

### Author Response · Authors · 2023-06-13
**Author feedback for all reviewers**

We thank all reviewers for their helpful comments and feedback. Our responses to each reviewer are provided below. We have also revised the manuscript per the reviewers’ suggestions, where we marked the changes in blue to help the reviewers find them more easily. Please find the list of changes we made in the *Changes Since Last Submission* section.